# The Relationship between Biofilm Phenotypes and Biofilm-Associated Genes in Food-Related *Listeria monocytogenes* Strains

**DOI:** 10.3390/microorganisms12071297

**Published:** 2024-06-26

**Authors:** Alexandra Burdová, Adriana Véghová, Jana Minarovičová, Hana Drahovská, Eva Kaclíková

**Affiliations:** 1Faculty of Natural Sciences, Comenius University, Ilkovičova 6, 84215 Bratislava, Slovakia; burdova23@uniba.sk (A.B.); hana.drahovska@uniba.sk (H.D.); 2Food Research Institute, National Agricultural and Food Centre, Priemyselná 4, 82475 Bratislava, Slovakia; adriana.veghova@nppc.sk (A.V.); jana.minarovicova@nppc.sk (J.M.)

**Keywords:** *Listeria monocytogenes*, biofilm formation, whole-genome sequencing, biofilm-associated genes

## Abstract

*Listeria monocytogenes* is an important pathogen responsible for listeriosis, a serious foodborne illness associated with high mortality rates. Therefore, *L. monocytogenes* is considered a challenge for the food industry due to the ability of some strains to persist in food-associated environments. Biofilm production is presumed to contribute to increased *L. monocytogenes* resistance and persistence. The aims of this study were to (1) assess the biofilm formation of *L. monocytogenes* isolates from a meat processing facility and sheep farm previously characterized and subjected to whole-genome sequencing and (2) perform a comparative genomic analysis to compare the biofilm formation and the presence of a known set of biofilm-associated genes and related resistance or persistence markers. Among the 37 *L. monocytogenes* isolates of 15 sequence types and four serogroups involved in this study, 14%, 62%, and 24% resulted in the formation of weak, moderate, and strong biofilm, respectively. Increased biofilm-forming ability was associated with the presence of the stress survival islet 1 (SSI-1), *inl*L, and the truncated *inl*A genes. Combining the phenotypic and genotypic data may contribute to understanding the relationships between biofilm-associated genes and *L. monocytogenes* biofilm-forming ability, enabling improvement in the control of this foodborne pathogen.

## 1. Introduction

*Listeria monocytogenes* is a foodborne pathogen that can cause listeriosis, a severe infection, especially in risk groups of pregnant women, the elderly, infants, and immunocompromised patients [1]. Despite its low incidence, listeriosis is associated with a high fatality rate [2]. The number of deaths from outbreaks in 2022, mainly caused by *L. monocytogenes*, was the highest ever reported in the EU in the last ten years. Listeriosis continues to be one of the foodborne infections with the highest number of hospitalizations and fatal cases in the EU. The overall EU case fatality rate in 2022 was 18.1% and a total of 35 outbreaks represented a 50% increase compared to 2021. Outbreaks mainly related to the consumption of ready-to-eat foods, such as cold smoked salmon, meat and meat products, dairy products, and frozen vegetables. In Slovakia, 25 confirmed human cases of invasive listeriosis were reported in 2022, corresponding to the highest notification rate (0.46 per 100,000 population) during the previous five years [3].

*L. monocytogenes* is a ubiquitous environmental bacterium, which can contaminate raw and processed food products at different production stages. Some strains of *L. monocytogenes* can persist even during food processing, thus increasing the likelihood of food product contamination [4,5]. The growth and survival of *L. monocytogenes* depend mainly on the ability to quickly adapt to changed conditions through a complex of stress factors [6]. Although the exact mechanism of persistence is still not elucidated, factors that can contribute to *L. monocytogenes* persistence in food environments include the ability to form biofilms and resistance to sanitizing agents, hygiene and sanitation processes, and refrigeration that suppresses the natural occurrence of less resistant competitive microflora [7,8,9].

Biofilm production of *L. monocytogenes* is presumed to be one of the ways that confer its increased resistance and persistence in the food chain [10,11]. Some *L. monocytogenes* strains may form strong biofilms on surfaces, which may contribute to their persistence for a long period in the production environment and therefore can become a permanent source of contamination [12,13]. The formation of biofilms can be affected by several factors, such as temperature, time, type of surface, and nutrient availability [14,15].

Several genetic mechanisms involved in biofilm formation in *L. monocytogenes* have been revealed [9]. *fla*A is one of the main biofilm-associated genes in *L. monocytogenes* encoding flagellin A, which appears to promote initial attachment [16]. The positive regulatory factor A gene, *prf*A, is involved in the later stages of biofilm development and also in virulence [17,18]. The *act*A gene encoding the actin-assembly-inducing peptide precursor is another important virulence determinant in *L. monocytogenes*, promoting bacterial aggregation and biofilm development [19]. The transcriptional regulator of stress response genes encoded by *sig*B is required for biofilm formation in the later stages of biofilm development [20]. Another biofilm-associated protein, encoded by *bap*L, appears to be involved in adherence in some *L. monocytogenes* strains; however, its role has not yet been clarified [16,21]. The virulence proteins internalins A and B have also been implicated in *L. monocytogenes* biofilm development, whereas *inl*A or *inl*B gene deletion in *L. monocytogenes* has been associated with a significant reduction in adherence [22]. Internalin L has been shown to play a similar role, as *inl*L deletion has been associated with reduced attachment [23]. The *agr*BDCA operon is a peptide-based quorum-sensing (QS) system in *L. monocytogenes* with an important role in biofilm development in the stage of adherence [24]. LuxS is another QS system that has been implicated in *L. monocytogenes* biofilm development [16], as mutations in *lux*S led to more readily attachment and production of denser biofilms [25]. Other genes implicated in *L. monocytogenes* biofilm development represent *rec*O and *lmo*2504, as both have been shown to be overexpressed in biofilm-associated cells in comparison to planktonic cells [26,27].

Some genes and complexes have been already indicated as associated with increased *L. monocytogenes* biofilm formation, adhesion capacity, and persistence abilities, in particular, stress survival islet 1 (SS-1) [7,28,29], the arsenite efflux transporter gene (*ars*D) [30], the internalin L gene (*inl*L) [7,23], truncated internalin A (*inl*A) [29,31], or *act*A genes [19]. An association between *L. monocytogenes* strains with *com*K prophage insertion and enhanced biofilm production has also been reported [32].

The ability of some *L. monocytogenes* strains to persist in food processing facilities for extended periods may be due to many factors. Alongside increased biofilm formation, tolerance to disinfectants such as quaternary ammonium compounds (e.g., *bcr*ABC, *qac*H, etc.), the presence of prophages and resistance markers on plasmids, and stress survival islands SSI-1 or SSI-2 are also important in a food production environment [33,34]. In this context, whole-genome sequence (WGS) analysis represents a powerful tool to reveal biofilm genotype–phenotype relationships, in terms of the diverse ability of *L. monocytogenes* strains to produce biofilms and potentially contribute to their persistence.

Understanding the genes involved in biofilm formation and their influence on biofilm structure will help identify new ways to eliminate harmful biofilms in food processing environments [35]. However, further surveys are needed to confirm the importance of certain genetic markers and to identify new ones [36].

The aims of the study were to (1) assess the biofilm formation of *L. monocytogenes* isolates from a meat processing facility and ewe’s milk farm previously characterized and subjected to whole-genome sequencing and (2) perform comparative genomic analysis for a comparison of biofilm phenotypes and genomes to identify genetic markers potentially associated with increased biofilm formation.

## 2. Materials and Methods

### 2.1. Bacterial Strains

Characteristics of 37 *L. monocytogenes* isolates used in this study are summarized in Table 1. The strains were isolated from the production chain of a meat processing facility in years 2011–2014 (20 isolates) and from ewe’s milk farm in years 2019–2021 (17 isolates). All strains were identified and characterized by molecular serogroup, PFGE and MLVA-typing, ST-MLST, and WGS-based cgMLST for persistence, as described in our previous studies [5,37,38].

*L. monocytogenes* ATCC BAA-679 (EGD-e strain), serotype 1/2a (American Type Culture Collection, Manassas, VA, USA), was used as a reference strain characterized by strong biofilm formation [24,39]. *L. monocytogenes* NCTC 11994, serotype 4b (National Collection of Type Cultures, RGU Aberdeen, Scotland, UK), was used as a reference strain characterized by weak/moderate formation [40].

The strains were kept in 20% glycerol or lyophilized for long-term storage at −18 °C in the Collection of Microorganisms, National Agricultural and Food Centre—Food Research Institute in Bratislava, Slovakia.

### 2.2. Quantification of Biofilm Formation

Quantification of biofilm formation was performed in tryptose soy broth (TSB; Merck, Darmstadt, Germany) in a microplate according to the previously described protocol [41], with minor modifications as follows: Briefly, cultures grown in TSB (for 18 h at 37 °C) were adjusted to obtain the optical density (OD) 0.2 at λ = 600 nm in a SmartSpec TM Plus spectrophotometer (Bio-Rad, Hercules, CA, USA) corresponding to approx. 10^6^ CFU/mL. Then, 200 μL volumes of these bacterial suspensions were added into each well of a sterile 96-well polystyrene microplate (Sarstedt, Nümbrecht, Germany). Negative control wells contained 200 μL of uninoculated TSB. The microplates were statically incubated for 24 h at 37 °C. The contents of each plate were discarded, and the biofilm was left to dry at laboratory temperature. The washing step using 150 μL of sterile phosphate-buffered saline (PBS; pH 7.3; Merck) added to each well was repeated three times. The biofilms were fixed by adding 150 µL of methanol to each well for 20 min, which was then discarded, and the biofilm was left to dry at laboratory temperature. For staining the bacterial biofilm, 150 μL of 1% *w*/*v* crystal violet (Loba Feinchemie, Fischamend, Austria) solution was added to each well and incubated statically for 20 min. After staining, the solution was removed by sharply tapping the plates upside down; the wells were washed three times with distilled water and completely air-dried. To quantify biofilm formation, 150 μL of 96% ethanol was added to dissolve the residual crystal violet, and after 10 min, the absorbance was measured at 600 nm using the Safire 2 Plate Reader (Tecan, Männedorf, Switzerland). Based on the results interpreted according to Stepanović et al. [41], the strains were classified as weak (NC-2xNC), moderate (2xNC-4xNC), or strong (>4xNC) biofilm formers, when the NC cutoff was calculated as mean of negative control wells + 3xSD. Each strain was tested in eight parallel wells in two independent assays, and the results were averaged. The results were processed with GraphPad Prism 5 (GraphPad Prism 5, San Diego, CA, USA).

### 2.3. Genomic Analysis

Whole-genome sequences of *L. monocytogenes* strains were obtained in studies aimed at their persistence in two different food processing environments [5,37,38]. Briefly, total bacterial DNA was used for preparing the sequencing library by Nextera XT and sequences were obtained on Illumina NextSeq or MiSeq systems. De novo assembly was performed by SPAdes [42] and contigs longer than 500 bp with coverage higher than 20 were annotated on BV-BRC (https://www.bv-brc.org, accessed on 16 May 2024). The seven loci MLST and specific gene content were determined using the *L. monocytogenes* MLST database (https://bigsdb.pasteur.fr/listeria/listeria.html, accessed on 10 May 2024) [43]. The presence of biofilm-associated genes and genome islands were checked in the MLST database, and 100% coverage and 90% similarity to known sequences were set as the limit of gene presence. Truncations in *inl*A were defined as present if a sequence was missing at least ten amino acids from the end of the sequence as compared to the EGD-e reference sequence.

## 3. Results

### 3.1. Biofilm Formation

The *L. monocytogenes* isolates exhibited varying levels of biofilm, from weak to strong production (Figure 1), when evaluated according to Stepanović et al. [41]. In particular, 14% (5/37), 62% (23/37), and 24% (9/37) of the analyzed strains resulted in weak, moderate, and strong biofilm formation, respectively.

The distribution of weak, moderate, and strong biofilm producers according to lineages, serogroups, and STs is shown in Figure 2. All weak producers, including NCTC11994, belonged to lineage I (serogroup IIb and IVb), while all strong producers, including EGD-e strain, belonged to lineage II (serogroups IIa and IIc) (Figure 2a,b). Strong biofilm producers were of ST9, ST20, ST121, ST394, and ST451 sequence types (Figure 2c).

### 3.2. Genome Analysis

The results of genome analysis are graphically summarized in Figure 3. The main biofilm-associated genes in *L. monocytogenes* (including those involved in virulence), in particular, *fla*A, *prf*A, *act*A, *inl*A, *inl*B, *sig*B, *agr*BDCA, *lux*S, *rec*O, and *lmo*2504, were found in all but one isolate (absence of *agr*B was observed). The *bap*L gene was found in 62% (23/37) of the isolates, all from lineage II, belonging to ST9, ST14, and ST121 and in the EGD-e strain.

Mutations leading to premature stop codons in *inl*A (truncated *inl*A) were identified in five (13.5%) isolates, which belonged to ST9 (three isolates) and ST121 (two isolates), and all of them were classified as strong biofilm producers.

The presence of the *inl*L gene was found in seven *L. monocytogenes* isolates belonging to lineage II; the majority of them were ST9 (three isolates) and ST394 (two isolates). Six of the seven *inl*L-positive isolates were classified as strong biofilm producers, the same as the EGD-e strain.

All analyzed strains were found to be positive for the *ars*D stress gene and contained the full version of the *act*A gene.

The stress survival islet 1 (SSI-1) was only present in four isolates, which belonged to lineage II, namely ST8 and ST9. The stress survival islet 2 (SSI-2) was exclusively found in ST121 isolates of lineage II. With the exception of one ST8 isolate, all these strains were strong biofilm producers.

The *bcr*ABC cassette was found in five ST14 *L. monocytogenes* isolates. *qac*H gene presence was exclusively associated with ST121 isolates possessing strong biofilm production.

## 4. Discussion

In this study, a comparative genomic analysis using WGS and a biofilm formation assay on the *L. monocytogenes* isolates collected from a meat processing facility and sheep farm in Slovakia were performed. The relationship between biofilm phenotypes and related genetic markers in 37 *L. monocytogenes* isolates and two collection strains were evaluated.

All the *L. monocytogenes* isolates analyzed in this study produced biofilms, but some of them formed significantly more biofilm than others. These results may suggest that due to their ability to form stronger biofilms in the food processing environment, some strains may have a competitive advantage over others. However, in relation to persistence, previous studies provided contradictory results on whether stronger biofilm formation is an indicator of persistence in processing environments [9,35,40].

Previous studies of relationships between lineage and biofilm formation also provided contradictory results, when Takahashi et al. [44] found lineage I strains to form more biofilm than lineage II, while Borucki et al. [45] and Combrouse et al. [46] presented opposite conclusions. Similar results were observed in this study, when all strong producers belonged to lineage II, while all weak producers belonged to lineage I.

Several studies have also suggested a relationship between biofilm production and *L. monocytogenes* serotypes. It was reported that serotype 4b strains formed higher levels of biofilm compared with serotype 1/2a strains [47]. However, opposite results were also achieved [48], and according to the ability to form a biofilm, *L. monocytogenes* serotypes were aligned in the order 1/2b, 1/2a, and 4b [28]. In our study, most of the IVb isolates (including the NCTC 11994 collection strain) were classified as weak biofilm producers. The *L. monocytogenes* isolates of the IIa serogroup, representing the largest portion of the analyzed isolates, showed moderate (21/27) or strong (6/27) biofilm-forming ability. *L. monocytogenes* IIc isolates, however, all belonged to the same ST9 and were classified as strong biofilm producers.

Several studies demonstrated that *L. monocytogenes* biofilm formation can be affected by various genes [9]. However, to date, only a few studies have been performed to identify *L. monocytogenes* biofilm-relevant genes on a genome-wide scale using WGS to reveal genetic factors that contribute to biofilm formation in food-related *L. monocytogenes* strains [36].

It was demonstrated in several studies that the biofilm is affected by the presence of the stress survival islet (SSI-1), which consists of five genes and is implicated in growth during exposure to stressful conditions in food environments [49]. It was shown that SSI-1 contributes to serotype-specific differences in biofilm formation in *L. monocytogenes* [29,30]. In this study, analysis of WGS data showed that SSI-1 was present in only 4 of the 30 *L. monocytogenes* isolates belonging to lineage II and in no isolate from lineage I. This finding is in agreement with the study of Painset et al. [50], where SSI-1 was over-represented in lineage II and absent in all lineage I isolates.

Considering the STs, SSI-1 was present in all strains from ST8 and ST9, which is in correlation with the results of Alvarez-Molina et al. [51] and DiCiccio et al. [36], while it was absent in other lineage II STs, including commonly widespread food-related *L. monocytogenes* STs 14, 121, and 451. The small number of isolates found to be SSI-1-positive in this study is limiting for confirming the association of SSI-1 presence with increased biofilm formation. However, three of the four isolates with this marker were assessed as strong biofilm formers. Stress survival islet 2 (SSI-2) was exclusively detected in strong biofilm-forming isolates belonging to ST121. This finding is in correlation with other studies [8,10,51].

The *ars*D gene was found in arsenic resistance operons of various bacteria [52] and has been associated with increased biofilm formation [30,36]. In this study, the stress gene *ars*D was present in all isolates.

Internalin proteins can affect *L. monocytogenes* biofilm formation, as well as adhesion, virulence, internalization into eukaryotic cells, and survival in the environment [53,54]. The *inl*A gene is a major virulence factor of *L. monocytogenes*, and truncations due to premature stop codons (PMSCs) caused virulence attenuation [55], while it led to significantly enhanced biofilm formation [31]. It has been found that lineage II strains carried *inl*A PMSC mutations more frequently than lineage I strains [56]. In our study, all *L. monocytogenes* strains belonging to ST9 and ST121 harbored truncated the *inl*A gene and were characterized by strong biofilm formation. It was also shown that CC9 and CC121 *L. monocytogenes* strains were frequently associated with food production sectors and hypovirulent in part due to truncations in the *inl*A gene [10,57].

Recently, it has been demonstrated that InlL contributes to the attachment of *L. monocytogenes* to abiotic surfaces and increased biofilm formation [7,23]. In our study, the *inl*L gene was found only in seven isolates belonging to five STs. Six of the eight *inl*L-positive isolates were classified as strong biofilm producers, as well as the EGD-e strain, which confirms the association between the presence of the *inl*L gene and increased biofilm formation. However, in the case of ST9 isolates, except for *inl*L, *inl*A truncation may also contribute to significantly increased biofilm formation (unpaired *t*-test; *p* < 0.05) in comparison to the rest of the strong biofilm formers.

Five strains were found to be positive for *bcr*A, *bcr*B, and *bcr*C genes, all of which belonged to ST14 isolates from the meat processing facility with increased tolerance to BAC and were considered persistent in our previous study [58]. However, all ST14 strains in this study (6 isolates from the meat plant and 11 isolates from the sheep farm) were characterized by moderate biofilm-forming ability regardless of the presence/absence of the *bcr*ABC cassette. Another multidrug resistance transporter gene, *qac*H, was found in two ST121 isolates only and was associated with biofilm, as both were classified as strong biofilm producers.

All the strong biofilm producers identified in this study belonged to ST9 (serogroup IIc), as well as ST20, ST121, ST394, and ST451 (all serogroup IIa). Among them, ST9 isolates showing the strongest biofilm-forming ability contained the *bap*L, SSI-1, *inl*L, and *inl*A truncation markers. In ST20, the presence of *bap*L and *inl*L was observed, while in ST394, the presence of *inl*L only, and in ST121, the presence of *bap*L, *inl*A truncation, and SSI-2 markers were found. The composition of biofilm-associated genes in these isolates was slightly different from that in the EGD-e reference strain, which possessed *bap*L, *inl*L, and SSI-1 and was also a strong biofilm producer, as shown previously [24,39].

Among the analyzed *L. monocytogenes* isolates, ST2, ST9 (from meat plant), and ST14 (from both facilities) were identified as persistent in our previous studies [5,37] based on less than 10 allelic differences in cgMLST of isolates present during more than one year period of sampling. However, no correlation between persistence and biofilm-forming ability was found, as ST2, ST9, and ST14 isolates were classified as weak, strong, and moderate biofilm producers, respectively.

*L. monocytogenes* clonal complexes CC9 and CC121 were frequently isolated from food and food processing environments [59]. CC121, as the most prevalent *L. monocytogenes* CC, followed by CC7, CC8, and CC9, were found in the Norwegian food system and were associated with an increased prevalence of stress survival and resistance determinants [60]. The analysis of fish production chains resulted in the predominant assignment of ST121 isolates from salmon [61]. A persistent ST451 strain was identified in a rabbit meat processing plant in the Czech Republic [62] and was found as the most prevalent CC, particularly in food and animal isolates during the 11-year study in Slovakia [63]. Recently, a large multi-country outbreak of invasive listeriosis by the *L. monocytogenes* ST394 clone linked to smoked rainbow trout was reported [64]. It can be assumed that these CCs/STs, classified as strong biofilm formers in our study, may be favored in terms of survival and subsequent dissemination in food-related environments.

## 5. Conclusions

In this study, the biofilm-forming ability of 37 food-related *L. monocytogenes* isolates collected from two different food processing environments (meat plant and sheep farm) in Slovakia was investigated. The genes potentially associated with biofilm formation were identified using WGS, as a faster and cheaper alternative technology to conventional typing methods. Based on the obtained results, it can be concluded that the presence of the SSI-1, *inl*L, and the truncated *inl*A genes, as well as their combination, was associated with increased biofilm-forming ability. The detection of genetic markers related to the biofilm formation of *L. monocytogenes* strains circulating in food processing environments may provide the opportunity to improve risk assessment for this important foodborne pathogen.

## Figures and Tables

**Figure 1 microorganisms-12-01297-f001:**
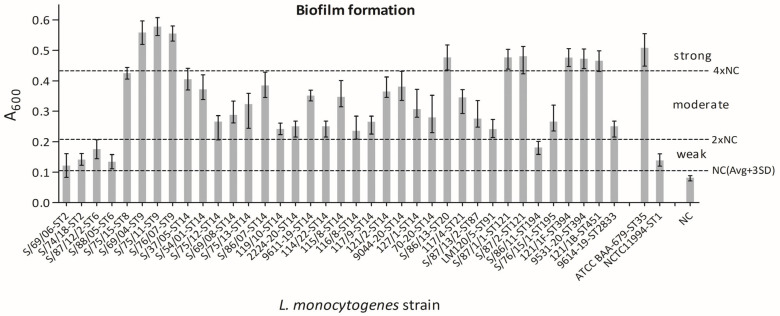
Levels of biofilm formation for individual *L. monocytogenes* strains. The isolates are ordered by MLST-ST. Data are the means ± standard deviation (SD) of eight parallel wells in two independent assays. The results were processed with GraphPad Prism 5.

**Figure 2 microorganisms-12-01297-f002:**
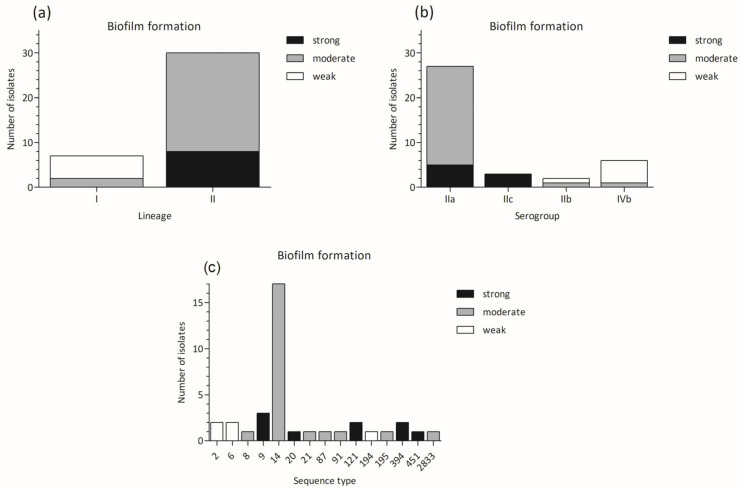
Biofilm production in *L. monocytogenes* strains belonging to different groups: (**a**) biofilm formation vs. lineages; (**b**) biofilm formation vs. serogroups; (**c**) biofilm formation vs. sequence types.

**Figure 3 microorganisms-12-01297-f003:**
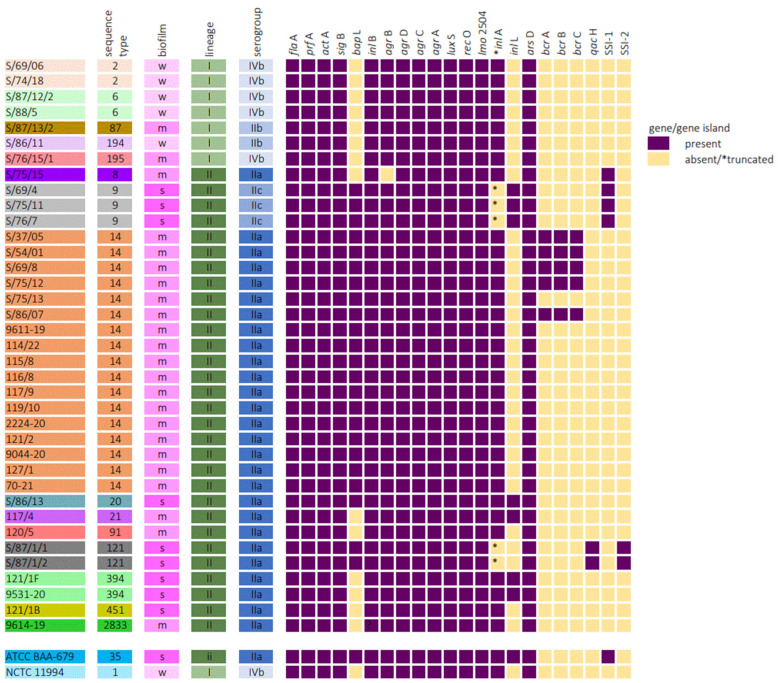
Association between biofilm production and presence/absence of biofilm-related genes in *Listeria monocytogenes* strains. In this study, 100% coverage and 90% similarity to known sequences were set as the limit of gene presence. Truncations in *inl*A were defined as present if a sequence was missing at least ten amino acids from the end of the sequence as compared to the EGD-e reference sequence.

**Table 1 microorganisms-12-01297-t001:** Characteristics of *L. monocytogenes* isolates and collection strains used in this study.

No.	Strain Designation	Date of Sampling	Source	Category	Sampling	Lineage	Serogroup	ST
1	S/37/05	2011-06	MP	NFC	washbasin	II	IIa	14
2	S/54/01	2012-05	MP	FC	grinder for raw meat	II	IIa	14
3	S/69/4	2013-07	MP	FC	chopping block	II	IIc	9
4	S/69/06	2013-07	MP	NFC	floor	I	IVb	2
5	S/69/8	2013-07	MP	FC	mixer for raw meat	II	IIa	14
6	S/74/18	2013-10	MP	NFC	floor	I	IVb	2
7	S/75/11	2014-02	MP	FC	grinder for raw meat	II	IIc	9
8	S/75/12	2014-02	MP	FC	mixer for raw meat	II	IIa	14
9	S/75/13	2014-02	MP	NFC	washbasin	II	IIa	14
10	S/75/15	2014-02	MP	FC	table	II	IIa	8
11	S/76/7	2014-04	MP	FC	grinder for raw meat	II	IIc	9
12	S/76/15/1	2014-04	MP	NFC	floor	I	IIb	195
13	S/86/07	2014-11	MP	RTE	final product	II	IIa	14
14	S/86/11	2014-11	MP	RTE	final product	I	IVb	194
15	S/86/13	2014-11	MP	NFC	floor	II	IIa	20
16	S/87/1/1	2014-12	MP	NFC	floor	II	IIa	121
17	S/87/2	2014-12	MP	NFC	floor	II	IIa	121
18	S/87/12/2	2014-12	MP	RTE	final product	I	IVb	6
19	S/87/13/2	2014-12	MP	RTE	final product	I	IIb	87
20	S/88/5	2014-12	MP	RP	raw pork meat	I	IVb	6
21	9611-19	2019-08	SF	RP	ewe’s milk	II	IIa	14
22	9614-19	2019-08	SF	RP	ewe’s milk	II	IIa	2833
23	114/22	2019-09	SF	RP	ewe’s milk	II	IIa	14
24	115/8	2019-09	SF	FC	milk filter	II	IIa	14
25	116/8	2019-09	SF	RP	ewe’s milk	II	IIa	14
26	117/4	2019-09	SF	FC	milk filter	II	IIa	21
27	117/9	2019-09	SF	RP	ewe’s milk	II	IIa	14
28	119/10	2019-12	SF	RP	ewe’s milk	II	IIa	14
29	120/5	2020-01	SF	FC	milk filter	II	IIa	91
30	2224-20	2020-03	SF	RP	ewe’s milk	II	IIa	14
31	121/1B	2020-07	SF	RP	ewe’s milk	II	IIa	451
32	121/1F	2020-07	SF	RP	ewe’s milk	II	IIa	394
33	121/2	2020-07	SF	RP	ewe’s milk	II	IIa	14
34	9044-20	2020-08	SF	RP	ewe’s milk	II	IIa	14
35	127/1	2020-09	SF	RP	ewe’s milk	II	IIa	14
36	9531-20	2020-09	SF	RP	ewe’s milk	II	IIa	394
37	70-21	2021-01	SF	RP	ewe’s milk	II	IIa	14
ATCC BAA-679		1924			rabbit tissue	II	IIa	35
NCTC 11994		1926			soft cheese	I	IVb	1

FC—food contact; NFC—non-food contact; RTE—ready to eat; RP—raw product; MP—meat plant, SF—sheep farm; ST—sequence type.

## Data Availability

The data underlying this article are available in the Listeria MLST database under accession numbers 81067-81089 and in GenBank NCBI at https://www.ncbi.nlm.nih.gov/genbank/, accessed on 10 May 2024 as Bioproject PR JNA897729. The data that support the findings of this study are available from the corresponding author upon reasonable request.

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
