# Peer review of "The Relationship between Biofilm Phenotypes and Biofilm-Associated Genes in Food-Related Listeria monocytogenes Strains"

_microorganisms, 2024, doi:10.3390/microorganisms12071297_

Round 1
Reviewer 1 Report
Comments and Suggestions for Authors
The manuscript submitted by Burdova aimed to study the relationship between biofilm phenotypes and biofilm-associated genes in food-related Listeria monocytogenes strains. Using different isolates, the biofilm formation was determined and the comparative genomic analysis was conducted to see the related genes. In my opinion, there were two major issues in the manuscript.
1.The comparative genome data was not deeply mined. It seems that the related genes identified based on the comparative genomic analysis was similar to the genes that have already been proofed to be involved in the biofilm formation. And the presentation of the result section is somewhat confused.
2.Just identification of the genes is not enough, experiments must be designed to show the logical clear relationship between the genes and the biofilm formation ability.
Author Response
Dear reviewer, thank you for reviewing our article sent to Microorganisms. I greatly appreciate your time and energy spent on revising in order to improve the manuscript from the position of your expertise.
1.The comparative genome data was not deeply mined. It seems that the related genes identified based on the comparative genomic analysis was similar to the genes that have already been proofed to be involved in the biofilm formation. And the presentation of the result section is somewhat confused.
Response: As we stated in the Abstract as well as in the Introduction sections, our goal was to compare the biofilm formation and the presence of known set of biofilm-associated genes and related resistance or persistence markers in food-related L. monocytogenes isolates. A special attention was paid to genes that were previously identified as potentially associated with increased biofilm formation in the studies, which were often limited on the analysis of individual strains and clinical isolates in most cases.
In the Results section, the obtained results are simply presented and graphically processed. Interpretation and comparison with published data are in the Discussion section, where the assessment is divided into two gene groups: genes associated with biofilm formation and genes associated with increased biofilm formation.
We would appreciate more detailed information which figures/text parts in the results section are confusing in order to correct these shortcomings. In order to improve the comprehensibility and facilitate the interpretation of the results, we modified Figure 3 by adding data directly to the table instead of the colour legend. Minor graphic corrections were also made in Fig. 1 and 2.
2. Just identification of the genes is not enough, experiments must be designed to show the logical clear relationship between the genes and the biofilm formation ability.
Response: Our study has been focused to presence/absence of genes previously identified as related to biofilm formation, and in particular, to increased biofilm formation. For this purpose, we used the WGS data obtained in our three previous studies aimed at tracing persistent L. monocytogenes contamination based on cgMLST and the subsequent determination of different levels of biofilm-forming ability of isolated strains. This study can be considered as a first step in revealing the relationships between biofilm production and related genes in a set of isolates from the food environment and the evaluation of the clear relationship between individual genes and the increased biofilm-forming ability could be due to the scale of the experimental work the subject for further studies. However, we consider the results of the study, the detection of genetic markers related to biofilm formation of L. monocytogenes strains circulating in food-processing environment, albeit limited to gene presence/absence, to be a meaningful contribution.
Reviewer 2 Report
Comments and Suggestions for Authors
The manuscript present important findings - biofilms in Listeria monocytogenes. There are some minor editing issues, and one more major issue. The authors repeatedly mention "significant associations" e.g. line 22 and present also a p-value in line 270. The authors also mention a statistics software two times, but they present only descriptive statistics and do not elaborate on which hypothesis tests they applied. This must be corrected.
Minor issues:
line 38: represent a 50 % increase
Table 1: "mill for raw meat" is a mincer or grinder?
line 257: .. while it led to a significantly enhanced biofilm production.."
Comments on the Quality of English Language
There are only minor issues.
Author Response
Dear reviewer, thank you for reviewing our article sent to Microorganisms. I greatly appreciate your time and energy spent on revising in order to improve the manuscript from the position of your expertise. I tried to do my best to improve the article based on your valuable comments. All changes in revised manuscript are highlighted in green.
There are some minor editing issues, and one more major issue. The authors repeatedly mention "significant associations" e.g. line 22 and present also a p-value in line 270. The authors also mention a statistics software two times, but they present only descriptive statistics and do not elaborate on which hypothesis tests they applied. This must be corrected.
Response:
The unjustified use of the term "significantly/significant", which was not supported by statistical analysis, was removed. (Lines 20-21, lines 312-314)
GraphPad Prism 5 was used to create graphs (Figure 1 and 2).
Unpaired t-test calculator, https://www.graphpad.com/quickcalcs/ttest1.cfm was used to evaluate statistical significance set as p<0.05, this information has been added (line 271)
Minor issues:
line 38: represent a 50 % increase
Table 1: "mill for raw meat" is a mincer or grinder?
line 257: ….. while it led to a significantly enhanced biofilm production."
Response:
Both suggested corrections in the text (line 37 and lines 258-259) were accepted.
The “mill for raw meat” was specified as grinder (Table 1)
Reviewer 3 Report
Comments and Suggestions for Authors
The authors of this study have tried to combine the biofilm-formation with the presence of biofilm-associated genes in food-related Listeria monocytogenes strains. The methods were well performed, whereas, the results are clearly described.
Comment
As it is shown in Figure 3, these STs (35,451,394,121,20 and 9) are strong-biolilm producers. The biofilm protects the microbial cells from the amtibiotics and from the difficult environmental conditions. So, I suppose that these clones have a avantage for survival and therefore dissemination. I propose to add a paragraph into discussion focused on the relationship between the predominance of some STs and their biofilm production. In addition, some epidemiological data from your country will streigth the study.
Author Response
Dear reviewer, thank you for reviewing our article sent to Microorganisms. I greatly appreciate your time and energy spent on revising in order to improve the manuscript from the position of your expertise. I tried to do my best to improve the article based on your valuable comments. All changes are highlighted in magenta
As it is shown in Figure 3, these STs (35,451,394,121,20 and 9) are strong-biolilm producers. The biofilm protects the microbial cells from the antibiotics and from the difficult environmental conditions. So, I suppose that these clones have a advantage for survival and therefore dissemination. I propose to add a paragraph into discussion focused on the relationship between the predominance of some STs and their biofilm production. In addition, some epidemiological data from your country will streigth the study.
Response: Several sentences on the predominant prevalence of L. monocytogenes CCs/STs in food and food-related environments, classified as strong biofilm producers in our study, were added at the end of the Discussion section (lines 294-305). Some epidemiological data from our country were added into the Introduction section (line 39-41).
Round 2
Reviewer 1 Report
Comments and Suggestions for Authors
The authors have addressed the questions well, and I have no further comments now.
Reviewer 3 Report
Comments and Suggestions for Authors
In this revised version all my suggestions were taken under consideration.